# DETECTING OUT-OF-DISTRIBUTION SAMPLES USING LOW-ORDER DEEP FEATURE STATISTICS

## ABSTRACT

The ability to detect when an input sample was not drawn from the training distribution is an important desirable property of deep neural networks. In this paper, we show that a simple ensembling of first and second order deep feature statistics can be exploited to effectively differentiate in-distribution and out-of-distribution samples. Specifically, we observe that the mean and standard deviation within feature maps differs greatly between in-distribution and out-of-distribution samples. Based on this observation, we propose a simple and efficient plug-and-play detection procedure that does not require re-training, pre-processing or changes to the model. The proposed method outperforms the state-of-the-art by a large margin in all standard benchmarking tasks, while being much simpler to implement and execute. Notably, our method improves the true negative rate from 39.6% to 95.3% when 95% of in-distribution (CIFAR-100) are correctly detected using a DenseNet and the out-of-distribution dataset is TinyImageNet resize. The source code of our method will be made publicly available.

## 1 INTRODUCTION

In the past few years, deep neural networks (DNNs) (Goodfellow et al., 2016) have settled as the state-of-the art-techniques in many difficult tasks in a plurality of domains, such as image classification (Krizhevsky et al., 2012), speech recognition (Hinton et al., 2012; Han et al., 2018), and machine translation (Bahdanau et al., 2014; van den Oord et al., 2017). This recent progress has been mainly due to their high accuracy and good generalization ability when dealing with real-world data. Unfortunately, DNNs are also highly confident when tested against unseen samples, even if the samples are vastly different from the ones employed during training (Hendrycks & Gimpel, 2017). Moreover, several works have shown that such deep networks are easily fooled by minor perturbations to the input (Goodfellow et al., 2014; Su et al., 2017). Obtaining a calibrated confidence score from a deep neural network is a problem under continuous investigation (Hendrycks & Gimpel, 2017) and a major thread in artificial intelligence (AI) safety (Amodei et al., 2016). In fact, knowing when the model is wrong or inaccurate has a direct impact in many production systems, such as self-driving cars, authentication and disease identification (Akhtar & Mian, 2018; Goswami et al., 2018), to name a few.

Guo et al. (2017) showed that despite producing significantly low classification errors, DNNs confidence scores are not faithful estimates of the true certainty. Their experiments confirmed that depth, width, weight decay, and batch normalization are the main reasons for overconfident scores. Moreover, they demonstrated that a simple and yet powerful method of temperature scaling in the softmax scores is an effective way to improve calibrate a DNNs confidence. While calibrating the classifier's output to represent a faithful likelihood from the training (**in-distribution**) data has effective solutions, the problem of detecting whether or not the samples are generated from a known distribution (**out-of-distribution**), is still an open problem (Hendrycks & Gimpel, 2017).

One straightforward approach to calibrate the classifier's confidence in order to detect samples whose distribution differs from the training samples distribution is to train a secondary classifier that digests both in-distribution (ID) and out-of-distribution (OOD) data so that anomalies are scored differently from ID samples, as performed in Hendrycks et al. (2018). Re-training a network, however, can be computationally intensive and may even be intractable, since the number of OOD samples is virtually infinite. Other solutions rely on training both classification and generative neural net-

Table 1: Summary comparison of the characteristics of recent related methods. Test complexity refers to the required number of passes over the network. Training data is the number of samples for which the methods were calibrated against (with all standing for the whole training set). AUROC is the area under receiver characteristic curve (detailed in Section 4). Performance shown is for DenseNet trained on CIFAR-100 and using TinyImageNet (resized) as OOD dataset.

| Method | Input pre-proc. | Model change | Test complexity | Training data | AUROC |
|---|---|---|---|---|---|
| Hendrycks & Gimpel (2017) | No | No | 1 | 1000 | 71.6 |
| Liang et al. (2018) | Yes | No | 3 | 1000 | 85.5 |
| Vyas et al. (2018) | Yes | Yes | 3 | all | 96.3 |
| Lee et al. (2018b) | Yes | Yes | 3 | all | 97.3 |
| Proposed | No | No | 1 | 1000 | **99.0** |

works for OOD using a multi-task loss (Lee et al., 2018a), or using unsupervised fully convolutional networks as done by Sabokrou et al. (2016) to detect OOD in video samples. All these methods, however, have a major drawback: they require re-training a modified model using a different loss function possibly with additional parameters, which increases the computational burden, and demands access to the entire original (and probably huge) training data.

In this work, we propose a novel OOD sample detection method that explores low-level statistics from feature layers. The statistics are obtained directly from the batch normalization layers (Ioffe & Szegedy, 2015), requiring no extra computations during training time, no changes to the network architecture and loss functions, nor preprocessing of the input image. During test time, the proposed method extracts statistics from selected layers and combines them into an OOD-ness score via a linear classifier. Throughout this paper, we observe that the mean and standard deviation of a given channel in a layer differ greatly between ID and OOD samples, which naturally motivates their use as features to be employed by an OOD detector. By selecting the BN layers of a network, we are able to normalize the features according to the learned batch statistics. The effectiveness of the proposed method is validated in two state-of-the-art DNN architectures (DenseNet and ResNet) (Huang et al., 2017; He et al., 2015; 2016; Zagoruyko & Komodakis, 2016) that are trained for image classification tasks in popular datasets. The proposed approach achieves state-of-the-art performance, surpassing all competitors by a large margin in all tested scenarios, while being much more efficient. Notably, our method only requires one forward pass while Liang et al. (2018); Lee et al. (2018b); Vyas et al. (2018) require two forward and one backward passes.

The rest of the paper is organized as follows. Section 2 describes prior work on OOD samples detection. Section 3 introduces the proposed method, whereas Section 4 details all experiments and compares the results with state-of-the-art methods. Finally, we draw our conclusions in Section 5.

## 2 PREVIOUS WORK

In this section, we describe recent prior work on OOD detection methods. A summary of all methods described can be seen in Table 1.

**Hendrycks & Gimpel (2017)** proposed a baseline method based on the posterior distribution (i.e. softmax scores). They showed that well-trained models tend to produce higher scores for ID samples than for OOD ones. Hence their method comprises of applying a threshold on the softmax-normalized output of a classifier. If the largest score is below the threshold, then the sample is considered OOD.

**Liang et al. (2018)** continued the aforementioned line of work and proposed the Out-of-Distribution detector for Neural networks (ODIN), which includes a temperature scaling $T \in \mathbb{R}_*^+$ to the softmax classifier as in Guo et al. (2017). The authors in ODIN argued that a good manipulation of $T$ eases the separation between in- and out-of-distribution samples. Allied to that, they also incorporated small perturbations to the input (inspired by Goodfellow et al. (2014)) whose goal is to increase the softmax score of the predicted class. Liang et al. (2018) found that this kind of perturbation has a stronger effect on ID samples than on OOD ones, increasing the separation between ID and OOD samples. ODIN outperforms the baseline method (Hendrycks & Gimpel, 2017) by a fair

margin; however, it is three times slower due to the two forward and one backward passes needed to preprocess the input, while (Hendrycks & Gimpel, 2017) only requires one forward pass.

**Vyas et al. (2018)** describes a novel loss function, called margin entropy loss, over the softmax output that attempts to increase the distance between ID and OOD samples. During training, they partition the training data itself into ID and OOD by assigning samples labeled as certain classes as OOD and use the different partitions to train an ensemble of classifiers that are then used to detect OOD samples during test time. They also use the input pre-processing step proposed in Liang et al. (2018), including temperature scaling.

**Lee et al. (2018b)** is the most recent work on OOD detection that we have knowledge of. They show that the posterior distribution defined by a generative classifier (under Gaussian discriminant analysis) is equivalent to that of the softmax classifier, and the generative classifier eases the separation between in- and out-of-distribution samples. The confidence score is defined using the Mahalanobis distance between the sample and the closest class-conditional Gaussian distribution. They argue that abnormal samples can be better characterized in the DNN feature space rather than the output space of softmax-based posterior distribution as done in previous work (e.g., ODIN). Samples are pre-processed similarly as done in ODIN, but the confidence score is increased instead of the softmax one. To further improve the performance, they also consider intermediate generative classifiers for all layers in the network. The final OOD sample detector is computed as an ensemble of confidence scores, chosen by training a logistic regression on validation samples. This method also shows remarkable results for detection of adversarial attacks and for incremental learning.

# 3 PROPOSED SOLUTION

An OOD detector should incorporate information from the training data in a natural manner, without being directly influenced by the loss function, which is intrinsically related to the task which could be well-defined for ID samples but be meaningless for most OOD samples. Moreover, if the OOD method is more dependent on the training distribution, it should be able to be applied to a wide variety networks, and not be designed specifically for a given architecture.

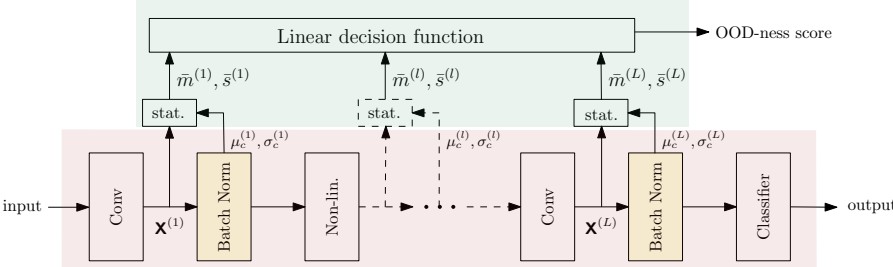

Figure 1: An illustration of the complete proposed model. At each BN layer, we extract the input, normalize it using the running statistics, and compute the first and second order features. The outputs are fed to a linear decision function to predict if the input sample is out-of-distribution or not.

Our method is based on a very simple observation. Input samples with different distributions generate statistically distinct feature spaces in a DNN. In other words, the deep features of an ID sample are distinct from those of an OOD one. Moreover, when using a normalization scheme, such as BN, the features are normalized by the statistics of the ID batch during training, possibly leading to feature statistics that are more similar to the batch statistics, as depicted in Figure 2.

The main problem then becomes how to summarize the feature space distribution for ID samples in a way that best discriminates between ID and OOD samples. In this work we show that using the first and second order statistics within each feature map performs remarkably well for this task.

Figure 1 shows the proposed OOD detector, highlighting the feature statistics that are extracted for each channel at each BN layer. These statistics are then combined through a learned linear decision function, providing a measure of OOD-ness. The way we compute the per-channel statistics

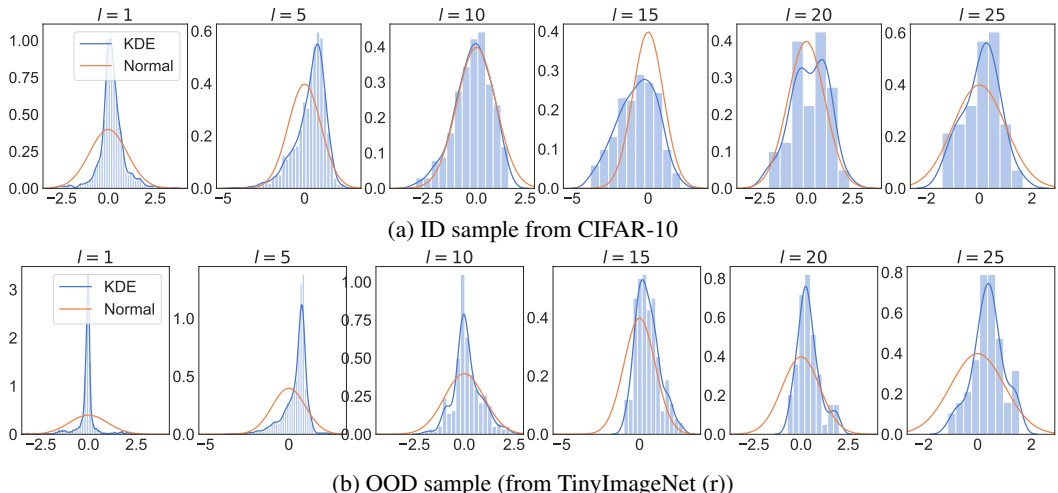

(a) ID sample from CIFAR-10

(b) OOD sample (from TinyImageNet (r))

Figure 2: Visualization of the normalized BN latent space distribution for one ID sample (a), and one OOD sample (b) at BN layers $l = 1, 5, 10, 15, 20, 25$ from a WideResNet-28-10. We also show the normal distribution for comparison. In the mid layers, the (arbitrary) ID CIFAR-10 sample (a) is more similar to the batch stastistics than the (arbitrary) OOD TinyImageNet(r) sample (b). Bars: normalized feature histograms. Blue curve: KDE. Orange curve: $\mathcal{N}(0, 1)$ (always the same inside each column but scaled for visualization). More in Appendix A.

is formalized in Section 3.1. Finally, the linear classifier used to combine the statistics from the different layers is described in Section 3.2.

## 3.1 LOW-ORDER FEATURE STATISTICS

In the previous section, we motivated that characterizing the feature-space distributions might lead to a robust OOD detector. As a first approach, one could model these distributions using a non-parametric method to estimate the distribution of the features for each channel, which requires the computation of sufficient statistics using the training data or using a parametric method to fit the distribution (Bishop, 2007), which are both computationally intensive.

Here, we propose to use only the mean and standard deviation computed along the spatial dimension for each channel to summarize the per-channel distribution. As it will be shown, these two statistics are sufficient to distinguish between ID and OOD. Moreover, because the mean and standard deviation of each channel are normalized by the running mean and variance computed during training by the BN layers (Ioffe & Szegedy, 2015), these statistics can be naturally combined within each layer to produce effective features for OOD detection. We describe such a procedure in what follows.

Given the $l$-th BN layer with input tensor $\mathbf{X} \in \mathbb{R}^{C \times W \times H}$, the output $\mathcal{BN}_{c,i,j}^{(l)}$ at channel $c$ is

$$Z_{c,i,j}^{(l)} = \frac{X_{c,i,j}^{(l)} - \mu_c^{(l)}}{\sqrt{[\sigma_c^{(l)}]^2 + \epsilon}}, \quad \mathcal{BN}_{c,i,j}^{(l)} = Z_{c,i,j}^{(l)} \gamma_c^{(l)} + \beta_c^{(l)}, \tag{1}$$

where $\gamma_c^{(l)} > 0$ and $\beta_c^{(l)}$ are the per channel per layer learned scaling and shifting parameters, $\epsilon > 0$ is a small constant value for numerical stability, $\mu_c^{(l)} \in \mathbb{R}$ and $[\sigma_c^{(l)}]^2 \in \mathbb{R}^+$ are the mean and variance estimated through a moving average using the batch statistics during training, and $\mathbf{Z}^{(l)} \in \mathbb{R}^{C \times W \times H}$ is the normalized feature tensor. It is worth noting that the statistics are calculated independently for each channel $c$ at each layer $l$.

In this paper, we conjecture that low-order statistics computed from either $X_{c,i,j}^{(l)}$ or $Z_{c,i,j}^{(l)}$ can be used to discriminate between ID and OOD samples. Given the unnormalized input $X_{c,i,j}^{(l)}$, we can

compute the empirical mean $m_c^{(l)}$ and standard deviation $\sigma_c^{(l)}$ features for each channel $c$ as

$$m_c^{(l)} = \frac{1}{WH} \sum_{i,j} X_{c,i,j}^{(l)}, \quad [s_c^{(l)}]^2 = \frac{1}{WH} \sum_{i,j} [X_{c,i,j}^{(l)}]^2 - [m_c^{(l)}]^2, \tag{2}$$

and the features for the normalized tensor $Z_{c,i,j}^{(l)}$ are defined as

$$
\begin{aligned}
\widehat{m}_c^{(l)} &= \frac{1}{WH} \sum_{i,j} Z_{c,i,j}^{(l)} = \frac{m_c^{(l)} - \mu_c^{(l)}}{\sigma_c^{(l)}} \\
[\widehat{s}_c^{(l)}]^2 &= \frac{1}{WH} \sum_{i,j} [Z_{c,i,j}^{(l)}]^2 - [\widehat{m}_c^{(l)}]^2 = \left[ \frac{\widehat{s}_c^{(l)}}{\sigma_c^{(l)}} \right]^2,
\end{aligned}
\tag{3}
$$

i.e., the normalized mean feature $\widehat{m}_c^{(l)}$ represents the difference between the sample mean $m_c^{(l)}$ and the BN running mean $\mu_c^{(l)}$ weighted by the running standard deviation $\sigma_c^{(l)}$, whereas $\widehat{s}_c^{(l)}$ is the normalized standard deviation by the BN running standard deviation $\sigma_c^{(l)}$.

### 3.2 Aggregation of feature statistics

**Intra-layers aggregation.** The features derived from low-order statistics (equation 3) can be readily used to train a predictor for ID/OOD discrimination. Of course, if they were produced for every feature map in the network, this would result in a feature vector of very high dimension, typically tens of thousands. Instead, we propose to combine these features within each BN layer, so that in the end we obtain one pair of features per layer: average mean and average variance. Taking advantage of the fact that features are approximately normalized by BN's running statistics, we propose to simply average them for all channels within a layer. Thus, each layer yields the following features, for the normalized case:

$$\bar{m}^{(l)} = \frac{1}{C_l} \sum_c \widehat{m}_c^{(l)}, \quad [\bar{s}^{(l)}]^2 = \frac{1}{C_l} \sum_c [\widehat{s}_c^{(l)}]^2, \tag{4}$$

where $C_l$ is the number of channels in layer $l$. Note that doing this aggregation amounts to computing the mean and standard deviation of all normalized features at the given layer. Using averages of the low-order statistics could lead to issues in deeper layers, where activations are in general concentrated over fewer number of channels. In this case, the mean of the statistics over channels might not be an appropriate data reduction function. Nevertheless, as we show in the experiments section, this did not impact the performance of the proposed method, but more investigation is warranted.

**Inter-layers ensemble and final classification.** Using all the features in equation 4, i.e., $f = (\bar{m}^{(1)}, \bar{s}^{(1)}, ..., \bar{m}^{(L)}, \bar{s}^{(L)})$ we fit a simple logistic regression model $h(f; \theta)$ with parameters $\theta \in \mathbb{R}^{2L+1}$. The parameters of the linear model are learned using a separate validation set formed with ID samples (positive examples) and OOD samples (negative examples).

## 4 Experiments

In this section, we evaluate the effectiveness of the proposed method in state-of-the-art deep neural architectures, such as DenseNet (Huang et al., 2017) and Wide ResNet (Zagoruyko & Komodakis, 2016), on several computer vision benchmark datasets: CIFAR (Krizhevsky, 2009), TinyImageNet, a subset of ImageNet (Deng et al., 2009), LSUN (Yu et al., 2015), and iSUN (Xu et al., 2015). We also use Gaussian noise and uniform noise as synthetic datasets. This evaluation protocol is the de facto standard in recent OOD detection literature (Hendrycks & Gimpel, 2017; Liang et al., 2018; Vyas et al., 2018; Lee et al., 2018b). All experiments were performed on four models trained from scratch (each one initialized with a different random seed) for each architecture, to account for variance in the model parameters. The code to reproduce all results is publicly available[1].

---

[1]Code for review avilable at `https://www.dropbox.com/s/o652ufpznatvx8u/ood-codes-review.zip?dl=0`.

## 4.1 SETUP

**Datasets:** For backbone training, we use CIFAR-10 and CIFAR-100 datasets which have 10 and 100 classes respectively, both containing 50,000 images in the training set and 10,000 images in the test set. At test time, the test images from CIFAR-10 (CIFAR-100) are considered as ID (positive) samples. For OOD (negative) datasets, we test with datasets containing natural images, such as TinyImageNet resize and crop, LSUN resize and crop, and iSUN, as well as synthetic datasets, such as Gaussian/uniform noise, which is the same dataset setup as in Liang et al. (2018); Hendrycks & Gimpel (2017). This is summarized in Table 2. For all datasets, we did the validation/test set split following the procedure in Liang et al. (2018). Just for reproducibility, 1000 samples from the test set are separated in a validation set used for fitting the logistic regressor and hyper-parameter tuning. The remaining samples (unseen for both backbone model and OOD detector) are used for testing.

Table 2: Summary of datasets. All generated samples have a size of $32 \times 32$.

|  |  | # samples val | # samples test | Pre-processing | Observation |
|---|---|---|---|---|---|
| ID | CIFAR-10 | 1000 | 9000 | - |  |
|  | CIFAR-100 | 1000 | 9000 | - |  |
| OOD | TinyImageNet (c) | 1000 | 9000 | Random patches | Subset of Imagenet containing 200 classes |
|  | TinyImageNet (r) | 1000 | 9000 | Downsampled |  |
|  | LSUN (c) | 1000 | 9000 | Random patches |  |
|  | LSUN (r) | 1000 | 9000 | Downsampled |  |
|  | iSUN | 1000 | 7925 | - | Subset of SUN images |
|  | Gaussian Noise | 1000 | 9000 | 0.5 mean and unit variance clipped in $[0, 1]$ | Channel-independent |
|  | Uniform noise | 1000 | 9000 | Uniform distribution in $[0, 1]$ |  |

**Backbone training:** Following Liang et al. (2018), we adopt the DenseNet (Huang et al., 2017) and Wide ResNet (Zagoruyko & Komodakis, 2016) architectures as our benchmark networks. For DenseNet, we use depth $L = 100$, growth rate $k = 12$, and zero dropout rate (DenseNet-BC-100-12). For Wide ResNet, we also follow Liang et al. (2018), with $L = 28$ and widen factor of 10 (WRN-28-10). All hyperparameters are identical to their original papers. All networks were trained using stochastic gradient descent with Nesterov momentum (Ruder, 2016) and an initial learning rate of 0.1. We train the DenseNet-BC-100-12 for 300 epochs, with batch size 64, momentum 0.9, weight decay of $10^{-4}$ and decay the learning rate by a factor of 10 after epochs 150 and 225. We train the WRN-28-10 for 200 epochs, with batch size 128, momentum 0.9, weight decay 0.0005, and decay the learning rate by a factor of 10 after epochs 60, 120, and 160. Table 3 shows each network error rate over 4 independent runs each one initialized with a different random seed.

Table 3: Average test error rates (%) and standard deviation (in parenthesis) over 4 runs.

| Architecture | CIFAR-10 | CIFAR-100 |
|---|---|---|
| DenseNet-BC-100-12 | 4.8 (0.2) | 22.6 (0.3) |
| WRN-28-10 | 4.0 (0.1) | 19.1 (0.3) |

**Logistic regression:** The logistic regressor is trained considering only the validation partitions for ID (positive examples) and OOD (negative examples) datasets (see Table 2). Using both mean and standard deviation as input (from equation 4), we have 50 features for WRN-28-10 models, and 198 features for DenseNet-BC-100-12 models. The training was performed using 5-fold cross-validation with the $\ell_2$ minimization and the regularization factor being chosen as the best one (according to the 5-folds) among 10 values linearly spaced in the range $10^{-4}$ and $10^4$.

**Evaluation metrics:** To evaluate the proposed method, we use the following metrics:

1. **True negative rate (TNR) at 95% true positive rate (TPR)**. Let TP, TN, FP, and FN be the true positive, true negative, false positive, and false negative, respectively. The TNR is defined as TN/(TN+FP) whereas TPR is defined as TP/(TP+FN).

2. **Area under the receiver operating characteristic curve (AUROC)**. AUROC is the area under the FPR=1-TNR against TPR curve.

## 4.2 MANIFOLD VISUALIZATION

We applied t-SNE (L. van der Maaten, 2008) to visualize our high-dimensional feature space in order to see the similarities between ID/OOD samples. For this, we used one of the WRN-28-10 models trained with CIFAR-10 as ID dataset. We fitted the t-SNE using the ID and all OOD validation samples together using both mean and standard deviation features, and the result is shown in Figure 3 using a perplexity of 30. It is clear from the visualization that the proposed features are concentrated around well-defined clusters. Both synthetic OOD datasets have clear distinct behavior from the natural images ones, and it is straightforward to differentiate them. TinyImageNet (c), LSUN (c) are similar and have some intersection with TinyImageNet (r), LSUN (r). Interestingly, the clustering seems to reflect the dataset generation method (resizing or cropping). Most importantly, one can see that the OOD samples are in different clusters than the ID (CIFAR-10) ones, which indicates that this feature choice is adequate for separating them.

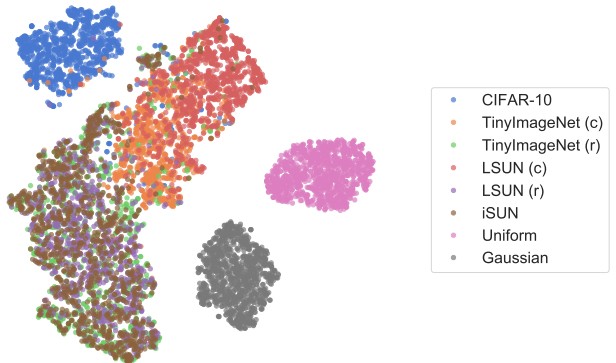

Figure 3: Manifold visualization of the proposed features (mean and std) from a WRN-28-10 model.

## 4.3 PAIRWISE FITTING

In this section, following other methods in the literature, we adjust the linear classifier for each ID/OOD pair. That is, for each pair of ID/OOD datasets, a different OOD classifier is trained using their respective validation samples.

We performed a comparison between the proposed method and four recent methods described in Section 2 (Hendrycks & Gimpel, 2017; Lee et al., 2018b; Liang et al., 2018; Vyas et al., 2018). Since Lee et al. (2018b) did not test using Wide ResNet models and the same datasets as in Liang et al. (2018); Vyas et al. (2018), here we only show the intersection between them: DenseNet BC 100-12 model, using CIFAR-10 (CIFAR-100) as in-distribution and TinyImageNet (resize) and LSUN (resize) as OOD distribution[2]. Extended results can be found in the appendix. For both Hendrycks & Gimpel (2017) and Liang et al. (2018) results, we reimplemented the method using their reference implementation[3,4]. For ODIN (Liang et al., 2018), we employed the same procedure as described by the authors to tune the methods parameters. A detailed description of the procedure can be found in Appendix B.1. For Lee et al. (2018b) and Vyas et al. (2018), we use the values presented on their papers (Table 2 and Table 3, respectively).

The results are compiled in Table 4. Notably, our method outperforms the baseline and ODIN methods by a large margin, and yields better results than Lee et al. (2018b) and Vyas et al. (2018) in all tested cases without requiring any preprocessing, or changing the backbone model. In fact, when setting the OOD-ness threshold to obtain 95% TPR, our method is able to correctly detect all OOD samples from the test partition.

---

[2]At the time of submission, Lee et al. (2018b) was still unpublished work

[3]Baseline: `github.com/hendrycks/error-detection`

[4]ODIN: `github.com/ShiyuLiang/odin-pytorch`

Table 4: Comparison between previous work (Section 2) and our proposed method using the TNR @ 95% TPR metric for DenseNet-BC-100-12. The displayed values (%) are the mean over 4 different training runs of the backbone models (for ODIN and our proposed model, the others results are from their respective papers).

| Dataset | OOD | Baseline | ODIN | Vyas et al. (2018) | Lee et al. (2018b) | Ours |
|---|---|---|---|---|---|---|
| CIFAR-10 | TinyImageNet (r) | 30.3 | 86.5 | 97.1 | 95.3 | 100.0 |
| | LSUN (r) | 35.5 | 92.7 | 99.2 | 97.5 | 100.0 |
| CIFAR-100 | TinyImageNet (r) | 21.0 | 52.9 | 79.5 | 86.6 | 100.0 |
| | LSUN (r) | 26.7 | 55.7 | 83.8 | 91.2 | 100.0 |

### 4.4 TESTING ON UNSEEN OOD DATASETS

We argue that the pairwise fitting scenario presented in the previous section is a limited performance measurement. In fact, many practical applications have OOD samples that do not come from a single distribution mode, and it might be infeasible to collect data from the many different modes of the OOD distribution (in general, some modes are unknown during training). A good OOD detector should be able to correctly identify samples from OOD distibutions for which its parameters have not been adjusted to. With this in mind, we propose a different, harder task in which an OOD detector is fitted to one, or a few, OOD datasets and then it is tested on all OOD datasets available. We note that this is not a standard practice in previous works, like in Liang et al. (2018). We begin by evaluating the generalization ability of our detector in some preliminary experiments, which motivate our decisions in the choice of OOD datasets for fitting the model and feature selection.

**Selecting Features:** We evaluate the individual impact of each of the proposed features (i.e., layers average mean and standard deviation) by comparing the performance of the classifier with different features as inputs. In this experiment, we fit the linear classifier to a specific OOD validation dataset and test on all of the OOD test datasets available. Our performance metric is the TNR @ 95% TPR averaged over all the tested datasets. Table 5 presents the performance of the classifier for the WRN-28-10, averaged of the four available models, with CIFAR-100 as ID dataset. The classifier fitted using only standard deviation features still achieved very good performance generalizing very well to unknown OOD datasets. Since we are interested in designing an OOD detector that is able to differentiate ID samples from any OOD sample, all our results from now on are presented using the averaged standard deviations per layer as the only features used to train the linear decision function.

Table 5: Generalization over unseen OOD datasets using different features (only mean, only std, and both). For each OOD validation set, we fit the linear model to it and tested against all OOD test sets. The results displayed are TNR @ 95% TPR averaged over the OOD datasets using WRN-28-10 as the backbone model and CIFAR-100 as ID dataset.

| Feat. | TinyImageNet (c) | TinyImageNet (r) | LSUN (c) | LSUN (r) | iSUN |
|---|---|---|---|---|---|
| $\bar{m}^{(l)}$ | 69.8 (31.2) | 81.6 (25.1) | 53.1 (39.7) | 79.1 (29.4) | 78.4 (29.2) |
| $\bar{s}^{(l)}$ | **98.5 (2.1)** | **93.6 (13.0)** | **87.5 (14.0)** | **88.3 (21.7)** | **87.3 (21.1)** |
| both | 90.7 (9.2) | 92.0 (13.6) | 70.3 (25.6) | 83.9 (27.2) | 84.4 (23.3) |

**Selecting OOD Dataset:** To understand to what extent a classifier trained considering one OOD dataset is able to generalize and detect samples from other OOD datasets, we trained the logistic regression considering as positive examples 1000 samples from the ID dataset (CIFAR-100) and as negative examples 1000 samples of a given OOD dataset, using the WRN-28-10 backbone model. As motivated in the previous section, we use the averages of normalized standard deviation features (equation 4). The obtained logistic regressor was then evaluated on the remaining OOD test datasets (unseen by both backbone training and logistic regressor). This procedure was then repeated for each possible OOD dataset, and the results are summarized in Table 6. We see that all classifier fitted using only natural images are capable to generalize well over all other OOD sets, while this is not entirely true when fitting on random noise datasets. Also, fitting to all OODs validation sets (penultimate row), we can achieve even higher TNR scores over all test sets.

**Using no OOD Dataset:** To further evaluate the effectiveness of the method, we also tested the extreme case where no OOD samples are available for training. To do this, we used an unsupervised

algorithm (one-class SVM (Manevitz & Yousef, 2002) with RBF kernel), and we only fitted to ID samples (i.e., no OOD samples are seen in the training step). The unsupervised results are summarized in the last row of Table 6. As one can see, even the unsupervised method shows reasonable performance; showing again that in the proposed feature space the ID/OOD samples have different behavior. This corroborates the assumption that these features are a good indicator of OOD-ness.

Table 6: Generalization to unseen OOD sets using CIFAR-100 as ID dataset and the WRN-28-10 backbone model. Performance of the OOD detector when the logistic regression is fit using 1000 samples of a given OOD dataset and then evaluated with respect to other OOD test datasets using only "std" as features. Results are TNR @ 95% TPR formatted as "mean (std)".

|  | | OOD test | | | | | | |
|---|---|---|---|---|---|---|---|---|
|  | | TinyImageNet (c) | TinyImageNet (r) | LSUN (c) | LSUN (r) | iSUN | Uniform | Gaussian |
| OOD fit | TinyImageNet (c) | 100.0 (0.0) | 96.9 (2.8) | 99.9 (0.0) | 98.2 (2.1) | 97.4 (2.6) | 99.8 (0.4) | 100.0 (0.0) |
|  | TinyImageNet (r) | 95.6 (3.1) | 99.8 (0.1) | 72.5 (17.6) | 100.0 (0.0) | 100.0 (0.0) | 99.3 (1.5) | 100.0 (0.0) |
|  | LSUN (c) | 99.9 (0.0) | 80.5 (11.5) | 100.0 (0.0) | 81.7 (10.8) | 79.9 (11.6) | 85.4 (29.1) | 100.0 (0.0) |
|  | LSUN (r) | 92.2 (3.6) | 99.7 (0.1) | 50.0 (21.2) | 100.0 (0.0) | 100.0 (0.0) | 97.0 (4.4) | 100.0 (0.0) |
|  | iSUN | 85.6 (4.3) | 99.7 (0.1) | 51.3 (20.7) | 100.0 (0.0) | 100.0 (0.0) | 78.8 (42.3) | 76.4 (46.9) |
|  | Uniform | 21.7 (2.3) | 73.8 (1.6) | 2.5 (0.4) | 81.2 (1.9) | 73.5 (1.9) | 100.0 (0.0) | 100.0 (0.0) |
|  | Gaussian | 15.5 (1.6) | 68.4 (1.7) | 1.5 (0.1) | 75.3 (2.4) | 67.5 (2.0) | 100.0 (0.0) | 100.0 (0.0) |
|  | **All** | 100.0 (0.0) | 99.7 (0.1) | 99.9 (0.1) | 100.0 (0.0) | 99.9 (0.0) | 100.0 (0.0) | 100.0 (0.0) |
|  | **Unsupervised** | 66.7 (4.1) | 88.3 (2.2) | 81.4 (1.2) | 92.8 (2.1) | 90.1 (2.3) | 100.0 (0.0) | 100.0 (0.0) |

**Comparison with ODIN (Liang et al., 2018):** We compare the generalization capabilities of our method with the state-of-the-art technique ODIN (Liang et al., 2018), in this new harder task. We fit both OOD classifier to maximize their detection performance on TinyImageNet (c) and Gaussian validation sets (i.e., 2000 OOD samples), and evaluate on all OOD test datasets. For our model, we use standard deviation features as inputs. For ODIN, we tune its hyperparameters using the grid search described in B.1. The results, presented in Table 7, show that our method outperforms ODIN by a large margin, indicating better generalization to samples from unseen OOD datasets.

**Effect of feature normalization:** We evaluated how much it helps to use the batch statistics computed by BN. As shown in Table 8, normalizing the latent space using the BN statistics before computing the features has clear advantages.

**Reduced validation data:** We study if our method can correctly detect OOD samples even when a small number of samples is available. Figure 4 shows the TNR @ 95% TPR for WRN 28-10 trained on CIFAR-10 (CIFAR-100), where only 27, 45, 75, 150, 300, 700, 1.5k, 3k (all) test images (ID + TinyImageNet crop + Gaussian, equally divided) are used to fit 25 coefficients of our logistic regressor. Using our method, only 27 images (from each ID and OOD), are necessary to achieve an average of 87.6% of TNR @ 95% TPR.

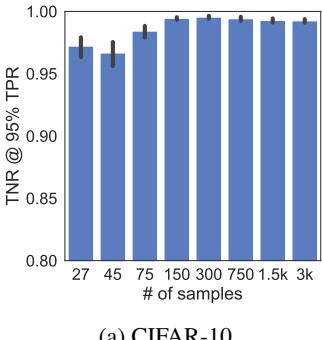
(a) CIFAR-10

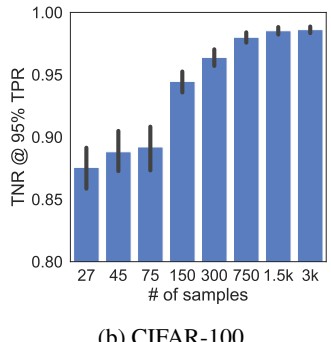
(b) CIFAR-100

Figure 4: Averaged TNR @ 95% TPR over the OOD datasets using only a few samples to fit the logistic regressor (for WRN-28-10). The logistic regressor was fitted using TinyImageNet (c) and Gaussian validation sets using only $\bar{s}^{(l)}$ as features. This experiment shows, in accordance to our results, that CIFAR-100 is more difficult to differentiate from other OOD datasets than CIFAR-10.

Table 7: Comparison between ODIN and our proposed OOD detector for several setups using image classification networks. All detector parameters (and ODIN's hyperparameters) were tuned for TinyImageNet (c) and Gaussian validation sets. The results are formatted as "mean (std)".

| | OOD dataset | TNR (95% TPR) | AUROC |
|---|---|---|---|
| | | **ODIN / Ours** | |
| **DenseNet BC 100-12** CIFAR-10 | TinyImageNet (c) | 90.5 (2.1) / **100.0 (0.0)** | 98.0 (0.3) / **100.0 (0.0)** |
| | TinyImageNet (r) | 76.9 (4.7) / **96.6 (1.4)** | 95.8 (0.8) / **99.3 (0.3)** |
| | LSUN (c) | 82.9 (6.1) / **100.0 (0.0)** | 97.0 (0.7) / **100.0 (0.0)** |
| | LSUN (r) | 86.2 (3.7) / **99.2 (0.5)** | 97.2 (0.6) / **99.8 (0.1)** |
| | iSUN | 81.1 (4.6) / **98.4 (0.9)** | 96.4 (0.8) / **99.6 (0.2)** |
| | Uniform | 68.7 (40.5) / **100.0 (0.0)** | 91.6 (9.2) / **100.0 (0.0)** |
| | Gaussian | 75.3 (31.2) / **100.0 (0.0)** | 96.2 (1.8) / **100.0 (0.0)** |
| **DenseNet BC 100-12** CIFAR-100 | TinyImageNet (c) | 72.9 (4.1) / **100.0 (0.0)** | 95.4 (0.4) / **100.0 (0.0)** |
| | TinyImageNet (r) | 39.6 (4.5) / **95.3 (0.7)** | 88.3 (1.3) / **99.0 (0.2)** |
| | LSUN (c) | 54.7 (4.7) / **100.0 (0.0)** | 93.3 (0.5) / **100.0 (0.0)** |
| | LSUN (r) | 42.2 (3.3) / **98.1 (0.7)** | 89.8 (0.8) / **99.5 (0.1)** |
| | iSUN | 34.4 (3.6) / **96.2 (1.0)** | 87.5 (0.9) / **99.2 (0.2)** |
| | Uniform | 0.0 (0.0) / **100.0 (0.0)** | 67.3 (7.9) / **100.0 (0.0)** |
| | Gaussian | 0.0 (0.0) / **100.0 (0.0)** | 39.0 (15.0) / **100.0 (0.0)** |
| **WRN 28-10** CIFAR-10 | TinyImageNet (c) | 91.9 (2.0) / **99.9 (0.0)** | 98.5 (0.3) / **100.0 (0.0)** |
| | TinyImageNet (r) | 73.4 (1.9) / **97.1 (1.3)** | 91.9 (0.8) / **99.4 (0.2)** |
| | LSUN (c) | 91.8 (2.3) / **100.0 (0.0)** | 98.4 (0.4) / **100.0 (0.0)** |
| | LSUN (r) | 84.7 (1.8) / **99.4 (0.4)** | 95.8 (0.4) / **99.8 (0.1)** |
| | iSUN | 79.8 (1.8) / **98.6 (0.9)** | 94.4 (0.5) / **99.6 (0.2)** |
| | Uniform | 98.9 (1.1) / **100.0 (0.0)** | 99.3 (0.3) / **100.0 (0.0)** |
| | Gaussian | **100.0 (0.0)** / **100.0 (0.0)** | 99.6 (0.3) / **100.0 (0.0)** |
| **WRN 28-10** CIFAR-100 | TinyImageNet (c) | 68.3 (3.0) / **100.0 (0.0)** | 94.6 (0.5) / **100.0 (0.0)** |
| | TinyImageNet (r) | 43.7 (2.3) / **96.8 (2.4)** | 87.7 (0.6) / **99.3 (0.5)** |
| | LSUN (c) | 38.8 (3.9) / **99.9 (0.0)** | 86.6 (2.3) / **99.9 (0.0)** |
| | LSUN (r) | 43.7 (1.3) / **98.2 (1.8)** | 88.5 (0.2) / **99.6 (0.4)** |
| | iSUN | 39.7 (2.0) / **97.5 (2.3)** | 87.2 (0.3) / **99.4 (0.4)** |
| | Uniform | 50.1 (49.0) / **99.8 (0.3)** | 78.5 (28.2) / **99.8 (0.3)** |
| | Gaussian | 25.3 (42.4) / **100.0 (0.0)** | 70.1 (26.2) / **100.0 (0.0)** |

Table 8: TNR @ 95% TPR for computing the averaged mean and standard deviation from unnormalized/normalized BN latent space. The backbone model is the WRN 28-10 using CIFAR-10 as ID samples, and the logistic regressor was fitted using TinyImageNet (c) and Gaussian validation sets. The results are formatted as "mean (std)".

| Normalization | TinyImageNet (c) | TinyImageNet (r) | LSUN (c) | LSUN (r) | iSUN |
|---|---|---|---|---|---|
| No | 99.8 (0.1) | 92.2 (1.0) | 99.8 (0.1) | 95.4 (0.6) | 93.2 (1.1) |
| Yes | **99.9 (0.0)** | **97.1 (1.3)** | **100.0 (0.0)** | **99.4 (0.4)** | **98.6 (0.9)** |

## 5 CONCLUSION

Deep neural networks trained to maximize classification performance in a given dataset are extremely adapted to said dataset. The statistics of activations throughout the network for samples from the training distribution (in-distribution) are remarkably stable. However, when a sample from a different distribution (out-of-distribution) is given to the network, its activation statistics depart greatly from those of in-distribution samples. Based on this observation, we propose a very simple yet efficient method to detect out-of-distribution samples. Our method is based on computing averages of low-order statistics at the batch normalization layers of the network, and then use them as features in a linear classifier. This procedure is much simpler and efficient than current state-of-the-art methods, and outperforms them by a large margin in the traditional ID/OOD fitting task (as proposed in previous works). We evaluated all methods in the challenging task of fitting on a single OOD dataset and testing on samples from other (unseen) datasets. In this harder scenario, our method generalizes well to unseen OOD datasets, outperforming ODIN by an even larger margin. Moreover, we show some preliminary results that even in the extreme case where no OOD samples are used for the training (unsupervised) we get reasonable performance.

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

# A SUPPLEMENTARY RESULTS FOR SECTION 3

(a) CIFAR-10

(b) TinyImageNet (c)

(c) TinyImageNet (r)

(d) iSUN

(e) Gaussian

Figure 5: More visualization results following the same procedure as in Figure 2 for a WRN-28-10 model.

# B SUPPLEMENTARY RESULTS FOR SECTION 4

## B.1 PARAMETER FINE-TUNING FOR ODIN

The temperature $T$ and noise magnitude $\epsilon$ hyperparameters were tuned using 1000 samples from the CIFAR test set as validation and evaluated on the remaining 9000 samples. The hyper-parameter

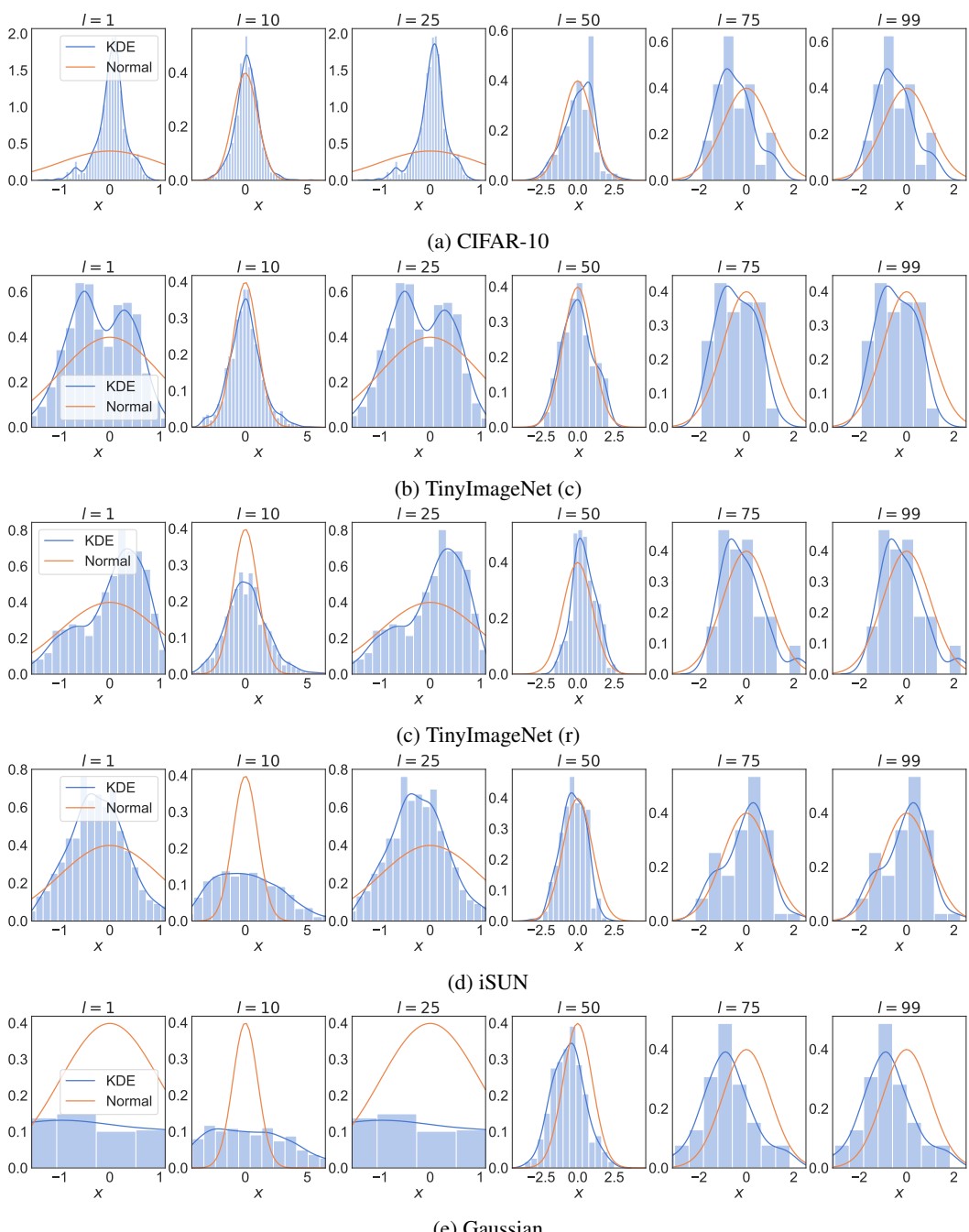

Figure 6: More visualization results following the same procedure as in Figure 2 but for a DenseNet-BC-100-12 model.

tuning is carried out on each pair of in- and out-of-distribution samples, which is the same procedure presented in Liang et al. (2018) and Lee et al. (2018b). For reproducibility, a grid search is employed considering $T \in \{1, 1000\}$ and $\epsilon$ with 21 linearly spaced values between 0 and 0.004 plus $[0.00005, 0.0005, 0.0011]$.

## B.2 ADDITIONAL BACKBONE MODEL

We also train a Wide ResNet with $L = 40$ and widen factor of 4 (WRN-40-4) using the same training setup in Section 4, and the results for both training on CIFAR-10 and CIFAR-100 is depicted in Table 9.

Table 9: Average test error rates (%) and standard deviation (in parenthesis) over 4 runs.

| Architecture | CIFAR-10 | CIFAR-100 |
|---|---|---|
| WRN-40-4 | 4.41 (0.12) | 20.83 (0.15) |

## B.3 OTHER EVALUATED METRICS

To compare all methods described in Section 2 to our proposed method, we also use the following additional metrics:

1. **Area under precision-recall curve (AUPR)**. The precision is evaluated as TP/(TP+FP) and recall in this case is the TPR. The AUPR-in (AUPR-out) is defined when in-(out-of)-distribution samples are considered as the positive ones.

## B.4 SELECTING OOD DATASET USING SVHN AS ID

We also tested using the Street View House Number (SVHN) dataset (Netzer et al., 2011) as ID and DenseNet BC 100-12 as backbone model. The pre-trained model is from Lee et al. (2018b), which has a test error rate of 3.58%. The results are displayed in Table 10.

Table 10: Generalization to unseen OOD sets using SVHN as ID dataset and the DenseNet BC 100-12 backbone model. Performance of the OOD detector when the logistic regression is fit using 1000 samples of a given OOD dataset and then evaluated with respect to other OOD test datasets using only "std" as features. Results are TNR @ 95% TPR.

| | | OOD test | | | | | | |
|---|---|---|---|---|---|---|---|---|
| | | TinyImageNet (c) | TinyImageNet (r) | LSUN (c) | LSUN (r) | iSUN | Uniform | Gaussian |
| **OOD fit** | TinyImageNet (c) | 100.0 | 100.0 | 100.0 | 100.0 | 100.0 | 100.0 | 100.0 |
| | TinyImageNet (r) | 100.0 | 100.0 | 100.0 | 100.0 | 100.0 | 100.0 | 100.0 |
| | LSUN (c) | 100.0 | 99.9 | 100.0 | 100.0 | 100.0 | 100.0 | 100.0 |
| | LSUN (r) | 100.0 | 100.0 | 100.0 | 100.0 | 100.0 | 100.0 | 100.0 |
| | iSUN | 100.0 | 100.0 | 100.0 | 100.0 | 100.0 | 100.0 | 100.0 |
| | Uniform | 100.0 | 99.7 | 100.0 | 100.0 | 99.9 | 100.0 | 100.0 |
| | Gaussian | 100.0 | 99.0 | 99.7 | 99.9 | 99.8 | 100.0 | 100.0 |
| | **All** | 100.0 | 100.0 | 100.0 | 100.0 | 100.0 | 100.0 | 100.0 |
| | **Unsupervised** | 88.0 | 88.8 | 81.6 | 95.0 | 91.9 | 100.0 | 100.0 |

## B.5 INTER-LAYER ENSEMBLE.

In this experiment, we evaluate the effect of ensembling across the BN layers, as described in Section 3.2, using a logistic regression. As shown in Figure 7, aggregating confidence scores of intermediate layers indeed improves the detection performance of OOD samples. The left-most bin indicates the TNR at 95% of TPR using only the last BN layer. As we go to the right, we aggregate the shallower BN layers, and the right-most bin indicates that all the BN layers are being used. Another important insight is that deeper layers (on the left of Figure 7) contribute more for the separation between in- and out-of-distribution samples.

## B.6 OOD DETECTION COMPARISON

We also extended our Table 7 with the extra backbone model (WRN 40-4) and additional metrics. The results are shown in Table 11.

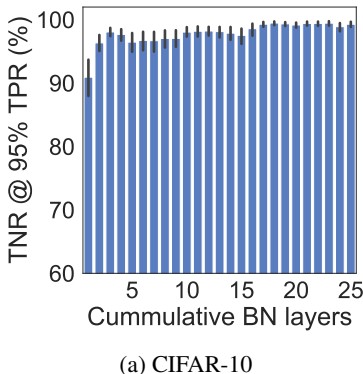

(a) CIFAR-10

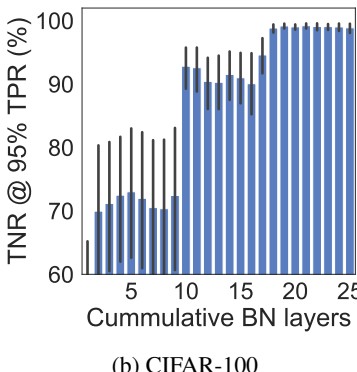

(b) CIFAR-100

Figure 7: TNR @ 95% TPR obtained when aggregating information from multiple layers. The leftmost bin corresponds to the deepest (last) layer, and the rightmost bin to the first BN layer in the network using a WRN-28-10 trained on CIFAR-10/CIFAR-100, the logistic regressor was fitted on TinyImageNet (c) + Gaussian validation sets and the results are an average over all OOD test datasets.

Table 11: Comparison between ODIN and the proposed OOD detector. The hyperparameters were tuned for TinyImageNet (c) and Gaussian validation sets and tested on others, and we only use the standard deviation as feature.

| | Out-of-distribution dataset | TNR (95% TPR) | AUROC | AUPR In | AUPR Out |
|---|---|---|---|---|---|
| | | ODIN / Ours | | | |
| **DenseNet BC 100-12** CIFAR-10 | TinyImageNet (c) | 90.5 (2.1) / **100.0 (0.0)** | 98.0 (0.3) / **100.0 (0.0)** | 98.5 (0.2) / **100.0 (0.0)** | 96.9 (0.6) / **100.0 (0.0)** |
| | TinyImageNet (r) | 76.9 (4.7) / **96.6 (1.4)** | 95.8 (0.8) / **99.3 (0.3)** | 96.4 (0.7) / **99.3 (0.3)** | 94.3 (1.2) / **99.3 (0.2)** |
| | LSUN (c) | 82.9 (6.1) / **100.0 (0.0)** | 97.0 (0.7) / **100.0 (0.0)** | 97.7 (0.5) / **100.0 (0.0)** | 95.7 (0.9) / **100.0 (0.0)** |
| | LSUN (r) | 86.2 (3.7) / **99.2 (0.5)** | 97.2 (0.6) / **99.8 (0.1)** | 97.8 (0.4) / **99.8 (0.1)** | 95.6 (1.1) / **99.7 (0.1)** |
| | iSUN | 81.1 (4.6) / **98.4 (0.9)** | 96.4 (0.8) / **99.6 (0.2)** | 97.3 (0.6) / **99.6 (0.1)** | 93.9 (1.6) / **99.5 (0.2)** |
| | Uniform | 68.7 (40.5) / **100.0 (0.0)** | 91.6 (9.2) / **100.0 (0.0)** | 95.0 (5.6) / **100.0 (0.0)** | 83.3 (13.1) / **100.0 (0.0)** |
| | Gaussian | 75.3 (31.2) / **100.0 (0.0)** | 96.2 (1.8) / **100.0 (0.0)** | 97.8 (1.1) / **100.0 (0.0)** | 89.2 (4.1) / **100.0 (0.0)** |
| **DenseNet BC 100-12** CIFAR-100 | TinyImageNet (c) | 72.9 (4.1) / **100.0 (0.0)** | 95.4 (0.4) / **100.0 (0.0)** | 96.5 (0.4) / **100.0 (0.0)** | 92.6 (0.4) / **100.0 (0.0)** |
| | TinyImageNet (r) | 39.6 (4.5) / **95.3 (0.7)** | 88.3 (1.3) / **99.0 (0.2)** | 89.8 (1.3) / **98.9 (0.2)** | 84.1 (1.3) / **99.0 (0.1)** |
| | LSUN (c) | 54.7 (4.7) / **100.0 (0.0)** | 93.3 (0.5) / **100.0 (0.0)** | 94.9 (0.4) / **100.0 (0.0)** | 90.2 (0.6) / **100.0 (0.0)** |
| | LSUN (r) | 42.2 (3.3) / **98.1 (0.7)** | 89.8 (0.8) / **99.5 (0.1)** | 91.6 (0.9) / **99.5 (0.1)** | 85.0 (0.8) / **99.5 (0.1)** |
| | iSUN | 34.4 (3.6) / **96.2 (1.0)** | 87.5 (0.9) / **99.2 (0.2)** | 90.4 (0.8) / **99.3 (0.2)** | 80.0 (1.1) / **99.1 (0.2)** |
| | Uniform | 0.0 (0.0) / **100.0 (0.0)** | 67.3 (7.9) / **100.0 (0.0)** | 79.7 (5.4) / **100.0 (0.0)** | 55.0 (5.5) / **99.9 (0.1)** |
| | Gaussian | 0.0 (0.0) / **100.0 (0.0)** | 39.0 (15.0) / **100.0 (0.0)** | 57.4 (13.2) / **100.0 (0.0)** | 41.2 (5.7) / **100.0 (0.0)** |
| **WRN 28-10** CIFAR-10 | TinyImageNet (c) | 91.9 (2.0) / **99.9 (0.0)** | 98.5 (0.3) / **100.0 (0.0)** | 98.8 (0.2) / **100.0 (0.0)** | 98.2 (0.4) / **100.0 (0.0)** |
| | TinyImageNet (r) | 73.4 (1.9) / **97.1 (1.3)** | 91.9 (0.8) / **99.4 (0.2)** | 89.0 (1.3) / **99.5 (0.2)** | 93.0 (0.6) / **99.4 (0.2)** |
| | LSUN (c) | 91.8 (2.3) / **100.0 (0.0)** | 98.4 (0.4) / **100.0 (0.0)** | 98.6 (0.3) / **100.0 (0.0)** | 97.9 (0.5) / **100.0 (0.0)** |
| | LSUN (r) | 84.7 (1.8) / **99.4 (0.4)** | 95.8 (0.4) / **99.8 (0.1)** | 94.6 (0.4) / **99.8 (0.1)** | 96.2 (0.5) / **99.8 (0.1)** |
| | iSUN | 79.8 (1.8) / **98.6 (0.9)** | 94.4 (0.5) / **99.6 (0.2)** | 93.4 (0.7) / **99.7 (0.1)** | 94.3 (0.6) / **99.6 (0.2)** |
| | Uniform | 98.9 (1.1) / **100.0 (0.0)** | 99.3 (0.3) / **100.0 (0.0)** | 99.5 (0.2) / **100.0 (0.0)** | 98.4 (0.4) / **100.0 (0.0)** |
| | Gaussian | **100.0 (0.0)** / **100.0 (0.0)** | 99.6 (0.3) / **100.0 (0.0)** | 99.8 (0.2) / **100.0 (0.0)** | 98.9 (1.0) / **100.0 (0.0)** |
| **WRN 28-10** CIFAR-100 | TinyImageNet (c) | 68.3 (3.0) / **100.0 (0.0)** | 94.6 (0.5) / **100.0 (0.0)** | 95.4 (0.5) / **100.0 (0.0)** | 93.7 (0.5) / **100.0 (0.0)** |
| | TinyImageNet (r) | 43.7 (2.3) / **96.8 (2.4)** | 87.7 (0.6) / **99.3 (0.5)** | 88.9 (0.7) / **99.3 (0.5)** | 85.9 (0.6) / **99.3 (0.5)** |
| | LSUN (c) | 38.8 (3.9) / **99.9 (0.0)** | 86.6 (2.3) / **99.9 (0.0)** | 88.6 (2.4) / **99.9 (0.0)** | 84.3 (2.2) / **99.9 (0.0)** |
| | LSUN (r) | 43.7 (1.3) / **98.2 (1.8)** | 88.5 (0.2) / **99.6 (0.4)** | 89.8 (0.2) / **99.6 (0.3)** | 86.3 (0.5) / **99.5 (0.4)** |
| | iSUN | 39.7 (2.0) / **97.5 (2.3)** | 87.2 (0.3) / **99.4 (0.4)** | 89.6 (0.3) / **99.5 (0.4)** | 83.1 (0.8) / **99.3 (0.5)** |
| | Uniform | 50.1 (49.0) / **99.8 (0.3)** | 78.5 (28.2) / **99.8 (0.3)** | 84.7 (20.7) / **99.9 (0.2)** | 76.7 (25.4) / **99.7 (0.5)** |
| | Gaussian | 25.3 (42.4) / **100.0 (0.0)** | 70.1 (26.2) / **100.0 (0.0)** | 79.6 (19.2) / **100.0 (0.0)** | 65.1 (21.9) / **100.0 (0.0)** |
| **WRN 40-4** CIFAR-10 | TinyImageNet (c) | 93.3 (1.1) / **100.0 (0.0)** | 98.3 (0.1) / **100.0 (0.0)** | 98.8 (0.1) / **100.0 (0.0)** | 97.4 (0.2) / **100.0 (0.0)** |
| | TinyImageNet (r) | 65.9 (4.0) / **96.9 (2.2)** | 89.7 (1.4) / **99.4 (0.3)** | 87.0 (1.7) / **99.4 (0.3)** | 90.1 (1.3) / **99.4 (0.3)** |
| | LSUN (c) | 93.0 (2.5) / **100.0 (0.0)** | 98.1 (0.3) / **100.0 (0.0)** | 98.6 (0.2) / **100.0 (0.0)** | 97.0 (0.6) / **100.0 (0.0)** |
| | LSUN (r) | 78.1 (2.2) / **98.6 (1.2)** | 93.9 (0.6) / **99.6 (0.2)** | 92.6 (0.8) / **99.7 (0.2)** | 93.4 (0.5) / **99.6 (0.2)** |
| | iSUN | 72.8 (3.0) / **97.9 (1.6)** | 92.5 (0.8) / **99.5 (0.3)** | 91.7 (0.9) / **99.6 (0.2)** | 91.0 (0.9) / **99.4 (0.3)** |
| | Uniform | 50.9 (32.3) / **100.0 (0.0)** | 84.5 (12.6) / **100.0 (0.0)** | 82.5 (14.6) / **100.0 (0.0)** | 81.4 (12.4) / **100.0 (0.0)** |
| | Gaussian | 55.8 (41.2) / **100.0 (0.0)** | 84.0 (15.2) / **100.0 (0.0)** | 82.8 (17.8) / **100.0 (0.0)** | 80.6 (14.2) / **100.0 (0.0)** |
| **WRN 40-4** CIFAR-100 | TinyImageNet (c) | 72.8 (2.2) / **100.0 (0.0)** | 95.2 (0.4) / **100.0 (0.0)** | 96.0 (0.4) / **100.0 (0.0)** | 93.4 (0.4) / **100.0 (0.0)** |
| | TinyImageNet (r) | 42.5 (2.6) / **95.2 (2.1)** | 85.8 (1.7) / **99.0 (0.4)** | 85.2 (2.4) / **99.0 (0.4)** | 83.8 (1.5) / **99.1 (0.3)** |
| | LSUN (c) | 44.3 (1.7) / **100.0 (0.0)** | 89.7 (0.7) / **99.9 (0.0)** | 91.7 (0.7) / **100.0 (0.0)** | 86.7 (0.9) / **99.9 (0.0)** |
| | LSUN (r) | 46.1 (2.6) / **98.0 (1.3)** | 87.8 (1.4) / **99.5 (0.2)** | 87.7 (2.0) / **99.5 (0.2)** | 85.2 (1.1) / **99.5 (0.2)** |
| | iSUN | 40.0 (3.0) / **96.9 (1.7)** | 85.8 (1.5) / **99.3 (0.3)** | 87.0 (1.8) / **99.4 (0.2)** | 81.1 (1.7) / **99.2 (0.3)** |
| | Uniform | 28.5 (39.1) / **99.7 (0.4)** | 71.3 (26.7) / **99.9 (0.2)** | 78.4 (19.7) / **99.9 (0.2)** | 68.1 (21.2) / **99.8 (0.4)** |
| | Gaussian | 5.3 (6.8) / **100.0 (0.0)** | 77.7 (11.8) / **100.0 (0.0)** | 84.5 (9.1) / **100.0 (0.0)** | 66.9 (10.7) / **100.0 (0.0)** |

## B.7 OOD DETECTION AT DIFFERENT TPR LEVELS

We also tested our proposed model on different TPR levels, and the results are depicted in Table 12.

Table 12: TNR at different TPR levels. The hyperparameters were tuned for TinyImageNet (c) and Gaussian validation sets and tested on others, and we only use the standard deviation as feature.

| | Out-of-distribution dataset | TNR (97% TPR) | TNR (99% TPR) |
|---|---|---|---|
| **DenseNet BC 100-12** CIFAR-10 | TinyImageNet (c) | 100.0 (0.0) | 100.0 (0.0) |
| | TinyImageNet (r) | 95.0 (2.2) | 88.8 (3.8) |
| | LSUN (c) | 100.0 (0.0) | 100.0 (0.0) |
| | LSUN (r) | 98.6 (1.0) | 94.9 (2.7) |
| | iSUN | 97.1 (1.5) | 91.5 (3.5) |
| | Uniform | 100.0 (0.0) | 100.0 (0.0) |
| | Gaussian | 100.0 (0.0) | 100.0 (0.0) |
| **DenseNet BC 100-12** CIFAR-100 | TinyImageNet (c) | 100.0 (0.0) | 100.0 (0.0) |
| | TinyImageNet (r) | 93.1 (0.8) | 85.2 (1.7) |
| | LSUN (c) | 100.0 (0.0) | 100.0 (0.0) |
| | LSUN (r) | 96.5 (1.3) | 89.4 (2.9) |
| | iSUN | 93.7 (1.6) | 84.8 (3.6) |
| | Uniform | 100.0 (0.0) | 100.0 (0.0) |
| | Gaussian | 100.0 (0.0) | 100.0 (0.0) |
| **WRN 28-10** CIFAR-10 | TinyImageNet (c) | 99.9 (0.0) | 99.8 (0.1) |
| | TinyImageNet (r) | 95.1 (2.1) | 89.3 (3.5) |
| | LSUN (c) | 100.0 (0.0) | 100.0 (0.1) |
| | LSUN (r) | 98.6 (0.9) | 95.7 (2.4) |
| | iSUN | 97.3 (1.6) | 92.3 (3.8) |
| | Uniform | 100.0 (0.0) | 100.0 (0.0) |
| | Gaussian | 100.0 (0.0) | 100.0 (0.0) |
| **WRN 28-10** CIFAR-100 | TinyImageNet (c) | 99.9 (0.1) | 99.7 (0.3) |
| | TinyImageNet (r) | 95.1 (4.0) | 89.1 (8.1) |
| | LSUN (c) | 99.9 (0.1) | 99.5 (0.5) |
| | LSUN (r) | 96.8 (3.3) | 92.1 (7.4) |
| | iSUN | 95.7 (4.1) | 89.6 (8.3) |
| | Uniform | 99.3 (1.4) | 95.5 (9.0) |
| | Gaussian | 100.0 (0.0) | 100.0 (0.0) |

