# OpenReview forum: "Detecting Out-Of-Distribution Samples Using Low-Order Deep Features Statistics"
_ICLR.cc/2019/Conference_

### Official Review · AnonReviewer1 · 2018-11-02
**A quick to implement and effective method for OOD sample detection.**

**Rating:** 7
**Confidence:** 4

**Review:**

The authors present a simple algorithm based on the statistics of neural activations of deep networks to detect out-of-distribution samples. The idea is to use the existing running estimate of mean and variance within BatchNorm layers to construct feature representations that are later fed into a simple linear classifier. The authors demonstrate superior performance over the previous state-of-art in the standard evaluation setting and provide fascinating insights and empirical analysis of their method.

There are several aspects of this work that I admire.

- The authors evaluate the generalization of their OOD detection model through evaluation against unseen OOD samples. This critical evaluation strategy is not typical in this literature and is much needed.
- The organization of the material and the depth of the discussion is of high quality. They discuss and connect the previous work, they clearly explain the idea and provide empirical results to support the design decisions, and run several experiments to evaluate their method from different angles followed by interesting discussions.
- The proposed method is easy to implement and has a minimal runtime complexity with no adverse effect on the underlying classifier.
- The source code is already included in the submission.

My only concern is that the feature pooling strategy first averages the input spatially, then across the channels. This feature size reduction is necessary because we have to ensure the following OOD classifier does not overfit in the validation stage. However, this reduction also introduces a permutation invariance in the feature space that is not desirable in OOD sample detection. I think it would make the work more valuable if the authors also take a critical look at the possible failure cases -- a short discussion of the weaknesses and assumptions.

Overall, the paper is technically sound and well-organized with sufficient coverage of the previous work. A thorough series of evaluations support the claims. It is a novel combination of existing techniques. The empirical evidence is strong and insightful. Given the simplicity of the method, I would expect a quick adoption by the community.
------
Rev. In light of the rebuttal and the following discussions I have updated my rating to 7.

---

> ### Author Response · Authors · 2018-11-19
> **Authors' response**
>
> We thank the reviewer for the detailed analysis of the method. Regarding your concern, we added a short note discussing  the implications of the feature reduction (pooling and averaging), the method assumptions and its weakness/limitations in Section 3.2:
>
> “Using averages of the low-order statistics could lead to issues in deeper layers, where activations are in general concentrated over a fewer number of channels. In this case, the mean of the statistics over channels might not be an appropriate data reduction function. Nevertheless, as we show in the experiments section, this effect has not impacted the performance of the proposed method,  but more investigation is warranted.”
>
> Bests,
> Authors.

---

### Official Review · AnonReviewer3 · 2018-11-02
**The paper suggests using Z-scores for comparing ID and OOD samples. Simple Idea, and much related work. Needs to address some issues.**

**Rating:** 5
**Confidence:** 4

**Review:**

There has been recent interest in using statistics and information summary measures to evaluate what deep nets are trying to do. Following the line of work, the paper suggests to use mean and variance of Z-scores accumulated across all layers/channels as features to distinguish ID and OOD samples. Simple idea but needs some work in its current format.

Firstly, the bulk of content in Sections 2 and 3 can be reduced/shortened since the importance of normalized statistics to understand learning models is well known, and not novel.

1) The choice of datasets/netowrks needs to be understood here. How is the OOD summary changing as more layers are added into computing the score (since the score is basically averaging all layers'/channels contribution)?
2)  What happens if we split the ID itself into two datasets and train on one, while using the other as OOD?
3)  (r) is random and (c) is not is it for the TinyImages? Seems to be the other way around.
4) What is the influence of the dataset? Since the summaries are first order statistics, there can be significant dependance of the 'coverage' of training data (i.e., how many and how good of instances are present for each class)? This is purely a sampling problem and it may reciprocate in the OOD scores (back to first order statistics). This needs to be tested.
5) Statistical tests of significance needs to be reported for the performance summaries shown in the Tables.

---

> ### Author Response · Authors · 2018-11-19
> **Authors' response**
>
> Thank you for your review.
>
> We revisited sections 2 and 3 to make sure they are as concise as possible.
>
> 1. The ID/OOD datasets, as well as the networks (WRN 28-10, DenseNet BC 100-12), were chosen to be able to compare our results with the state-of-the-art. Figure 7 (in the Appendix) shows how the TNR changes as more layers are added to the logistic regressor. For the WRN 28-10,  using only ⅓ of the layers gave us already more than 90% of TNR @ 95% of TPR, and adding more layers (rightmost bins) is really helpful to the OOD-ness score.
>
> 2.  In our experiments, we always split the ID dataset between training (used for training the backbone), validation (used for training the logistic regressor) and test partitions (used for testing our OOD detector, unseen by both backbone and logistic regressor), and the samples from the ID dataset are correctly classified as such (see Table 6, 7, 11). We also tested training the logistic regressor with arbitrary samples from the training dataset (Reviewer 2, Q8 answer), and the results did not change, indicating that the features do not change significantly between the partitions. Also, the tSNE plot (Figure 3) shows that ID/OOD samples are different in our proposed feature space. We hope this answers your doubt.
>
> 3. (r) stands for resize, while (c) stands for crop.
>
> 4. To address your concern,  we now show on new Table 6 the results of using the summaries from one dataset to evaluate on another one.
>
> 5. We trained 4 models for each network architecture (each one initialized with a different random seed) and all results reported consider the standard deviation of the results (through averaging, median, and other relevant statistics). In the revised manuscript, we specify the notation in each table. We would like to note that none of the previous works reported results considering different realizations of each backbone model.
>
> Bests,
> Authors.

---

### Official Review · AnonReviewer2 · 2018-11-05
**Appealing results but presentation lacks clarity and correctness of experiment is challenged.**

**Rating:** 5
**Confidence:** 4

**Review:**

Summary: A relatively simple approach for detecting out-of-distribution samples by having a parallel logistic regression model using simple statistics (mean and variance) over output of each batch normalisation layer, in order to discriminate between in-distribution and out-of-distribution samples. Results are appealing but presentation is lacking clarity at time and some doubts on the correctness of the experiments remain.

With the goal of detecting out-of-distribution sets, the authors propose to use logistic regression over simple statistics (mean and variance) of each batch normalization layer of CNN in order to discriminate between in-distribution (ID) and out-of-distribution (OOD) samples. They argue that ID and OOD samples can be discriminated with these statistics.

Quality: The motivations of the paper are clear, it aims at having better capacity to detect OOD samples with a method that involves less computations. However, the quality of the experiments is not good enough and I have doubts on their validity.

Originality: Ok. The proposal is relatively simple and is based on the intuition that statistics for the batch normalization is useful to detect OOD samples. The problem is not new, the approach is relatively ad hoc, but it works.

Significance of the work: The results reported are unreasonably good. Although the authors claim the improvement of detection of OOD is significant, the results achieved by detecting **all** the out-distribution samples sounds weird and irrational. How rejecting all Tiny-ImageNet is possible while there are several overlap between the classes presented in TinyImageNet vs. Cifar10/Cifar100 (cf. Table 8)? To me, it looks like the model either overfitted something else than the content of the images, maybe the background noise or similar regarding the nature of the data. More experiments with different in-distribution datasets should be made to be convincing. All experiments reported are using either Cifar10 or Cifar100 as in-distribution datasets.

The author also claim using few samples from a single OOD set is enough for training the regressor that provides OOD-ness score. Is it true for any OOD set or only a carefully chosen OOD set can demonstrate this behavior? What is the criteria for selecting a good OOD set for training the regressor?

Despite of the fact that the proposed method is heavily dependent on the threshold, the authors barely discuss of it. I am assuming that threshold is on OOD-ness score, is that correct? How does look like the OOD-ness score for an ID set over different OOD sets? Providing the OOD-ness score for ID and OOD could reflect how the proposed method is sensitive to a selected threshold. In other words, is selecting a fix threshold will to the TPR / FPR across different OODs.

The overall writing style is perfectible. I did not found the paper super clear in the presentation and it is difficult to really get all useful information for it. However, the authors appear knowledgeable of the literature and the overall structure is clear.

An example of lack of clarity in the explanations: in Table 7, I have difficulty to make sense of the 100% achieved for “Ours (pair)” vs “Ours”. Is the “Ours (pair)” the rate obtained with the exact pair used for adjusting the threshold, while “Ours” is on another dataset? If not, what this mean? Moreover, reporting columns all with 100% is not a good practice, it seems to be a stunt to impress the reader, while not carrying much in term of content and understanding.

In Table 8, I do not understand what the values in parenthesis means.

Another element: why for training the regressor, the IN and OOD samples are not selected from their corresponding training sets instead of splitting their test sets to a validation and test sets?

---

> ### Author Response · Authors · 2018-11-19
> **Authors' response**
>
> [Part 2/2]
>
> The choice of OOD dataset for this experiment was done based on the performance on separate validation samples. In order to highlight the impact of selecting a given OOD dataset for training the logistic regressor, we’ve added Table 6 to the manuscript.  Note that this is an advantage w.r.t. the published SOTA (ODIN [2]), where they fit the parameters to the same OOD dataset they use to test, whereas our method generalizes to unseen OOD datasets. To address your concern even further, we performed unsupervised training of our model using one-class SVM, meaning no OOD samples are used during training time, only samples from the ID distribution. As one can see in the new results added to Table 4, the method is still able to separate the ID samples with reasonable performance without ever seeing any OOD sample during training.
>
> Q4) “Despite of the fact that the proposed method is heavily dependent on the threshold, the authors barely discuss of it. I am assuming that threshold is on OOD-ness score, is that correct? How does look like the OOD-ness score for an ID set over different OOD sets? Providing the OOD-ness score for ID and OOD could reflect how the proposed method is sensitive to a selected threshold. In other words, is selecting a fix threshold will to the TPR / FPR across different OODs.”
>
> The chosen threshold is always set to obtain 95% TPR in the dataset used to train the logistic regressor. We do not further adjust the threshold for each OOD.  We thank the reviewer for pointing this out. We realized this was not clear in the manuscript, and we amended the manuscript to clarify this point. It should also be pointed out that the AUROC value, shown in Table 7, provides a metric that does not depend on selecting a threshold.
>
> Q5) “An example of lack of clarity in the explanations: in Table 7, I have difficulty to make sense of the 100% achieved for “Ours (pair)” vs “Ours”. Is the “Ours (pair)” the rate obtained with the exact pair used for adjusting the threshold, while “Ours” is on another dataset? If not, what this mean?“
>
> We agree that this was misleading in the submitted manuscript. We reorganized the experimental section to make it clearer. For clarification, “Ours (pair)” mean that we fitted the linear classifier in a validation partition of the OOD validation set and tested in another partition of the  OOD, while “Ours” means that we fitted both the parameters and the threshold to one OOD validation set (the combined validation partitions of TinyImageNet (c) and Gaussian noise) and tested on all other  OOD dataset’s test partition.
>
> Q6) “Moreover, reporting columns all with 100% is not a good practice, it seems to be a stunt to impress the reader, while not carrying much in term of content and understanding.”
>
> The results in this table follow the standard protocol in the OOD detection literature, by setting the threshold to 95% TPR [Hendrycks2017, Liang2018,Lee2018a, Lee2018b]. We added an explicit comment explaining this. In other words, we are only picking one point in the ROC curve that. Please refer to our answer to Q1 where we show that the 100% values stem from this arbitrary threshold.
>
> Q7) "In Table 8, I do not understand what the values in parenthesis means."
>
> Thanks for pointing this out. In fact, we realized we this wasn't enough stressed in the manuscript. . They correspond to the standard deviation after running each experiment four times (each one on a different backbone model trained from scratch on the ID dataset) to understand how the results varied across different trained models. Note that this is not common in literature where only one run is reported.
>
> Q8) “Another element: why for training the regressor, the IN and OOD samples are not selected from their corresponding training sets instead of splitting their test sets to a validation and test sets?”
>
> To do a fair comparison, we follow the same data split as previous works.  Some experiments that we did internally indicate that the feature distribution of training and validation samples is essentially the same. Just to state some numbers, for a DenseNet BC 100-12 model using CIFAR-100 as ID model, we trained the logistic regressor using 1000 arbitrary samples from the training set (i.e., 1000 samples chosen randomly over 50000 images) + 2000 validation samples from OOD datasets (TinyImageNet crop + Gaussian). We ran this experiments 5 times and we got the results displayed below as mean (std) calculated over the runs:
>
> OOD dataset      | TNR @ 95 TPR
> TinyImageNet (c) | 100.0 (0.0)
> TinyImageNet (r) | 93.6 (2.9)
> LSUN (c)         | 100.0 (0.0)
> LSUN (r)         | 97.8 (2.1)
> iSUN             | 95.9 (2.8)
> Uniform          | 100.0 (0.0)
> Gaussian         | 100.0 (0.0)
>
> Which is a similar result (given the variance) to the one we reported in Table 7 (using the 1000 validation samples).
>
> Best,
> The authors.

---

> ### Author Response · Authors · 2018-11-19
> **Authors' response**
>
> [Part 1/2]
> Thank you very much for your detailed review.  We have re-organized the experimental section to overcome the clarity issue raised by the reviewer. With our answers below and the improvements on the overall manuscript presentation, we hope we have addressed all the concerns raised.
>
> Before diving into the detailed answers, we would like to stress the following two important points:
>
> i)  We follow the experimental framework used in the OOD detection literature: Liang et al. (ICLR 2018),  Lee et al. (ICLR 2018), Vyas et al. (ECCV 2018), Lee et al. (to appear in NIPS 2018) . This was imperative to present a fair comparison with the recent literature. We also go beyond this protocol by testing generalization to unseen OOD datasets.
> ii) The OOD-ness score threshold is always set to achieve 95% TPR following the benchmark metric commonly used in the literature for evaluating OOD methods. This means that there is always 5% of ID samples that will be  classified as OOD, so the separation is never perfect. In our first detailed answer below, we show results when varying this 95% threshold.
>
> Q1) “Although the authors claim the improvement of detection of OOD is significant, the results achieved by detecting **all** the out-distribution samples sounds weird and irrational. How rejecting all Tiny-ImageNet is possible while there are several overlap between the classes presented in TinyImageNet vs. Cifar10/Cifar100 (cf. Table 8)?"
>
> As we previously mentioned, by setting the threshold to achieve 95% TPR, we are also rejecting 5% of ID samples. The 100%  value for this experiment is superior but not unreasonably far from state-of-the-art methods.  If we increase the TPR level for CIFAR-100 as ID and using the WRN 28-10 architecture (fitted only to TinyImageNet crop + Gaussian valid samples), we have:
>
> OOD dataset      | TNR @ 95% TPR | TNR @ 97% TPR  | TNR @ 99% TPR
> TinyImageNet (c) | 100.0 (0.0)   | 99.9  (0.1)    | 99.7  (0.3)
> TinyImageNet (r) | 96.8  (2.4)   | 95.1  (4.0)    | 89.1  (8.1)
> LSUN (c)         | 99.9  (0.0)   | 99.9  (0.1)    | 99.5  (0.5)
> LSUN (r)         | 98.2  (1.8)   | 96.8  (3.3)    | 92.1  (7.4)
> iSUN             | 97.5  (2.3)   | 95.7  (4.1)    | 89.6  (8.3)
> Uniform          | 100.0 (0.0)   | 99.3  (1.4)    | 95.5  (9.0)
> Gaussian         | 100.0 (0.0)   | 100.0 (0.0)    | 100.0 (0.0)
>
> We added a more detailed table with different TPR levels in the Appendix (Table 12) to stress this point.
> We do believe that the semantic overlap does merit a more thorough investigation, especially looking into what constitutes an OOD sample. For instance, if it is a sample belonging to a training class but in a different dataset (e.g. CIFAR vs TinyImageNet), or if it is a sample from the same dataset but of a different class (e.g. splitting CIFAR). In this work, following previous approaches, we concentrate on the first case.
>
> Q2) “To me, it looks like the model either overfitted something else than the content of the images, maybe the background noise or similar regarding the nature of the data. More experiments with different in-distribution datasets should be made to be convincing. All experiments reported are using either Cifar10 or Cifar100 as in-distribution datasets.”
>
> We agree that we do not know precisely what underlying factors of the data the method is adjusting to, which is a common issue to all deep learning approaches. On the other hand, the features learned by the network are adapted to the task it was trained for (classification). We use the batch statistics computed over these features to adjust a small number of parameters in the linear classifier. Again, the choice of CIFAR-10 and CIFAR-100 as ID dataset is standard in the literature [Hendrycks2017, Liang2018].  We also added an experiment in the Appendix (Table 10) showing the results for a different ID dataset, SVHN, with similar performance than was obtained for the CIFAR datasets.
>
> Q3) “The author also claim using few samples from a single OOD set is enough for training the regressor that provides OOD-ness score. Is it true for any OOD set or only a carefully chosen OOD set can demonstrate this behavior? What is the criteria for selecting a good OOD set for training the regressor?”

---

> > ### Public Comment · (anonymous) · 2018-11-24
> > **Brief Comment on Tuning on OOD Samples**
> >
> > "We follow the experimental framework used in the OOD detection literature... We also go beyond this protocol by testing generalization to unseen OOD datasets."
> > The original Hendrycks et al. 2017 does not assume access to samples for tuning from the OOD test sets. Liang et al. 2018 made this assumption, but they did not defend why they made that assumption.
> >
> > Your use of SVHN, CIFAR-10, CIFAR-100 seems fine.
> >
> > It is customary to train Wide ResNets (but not ResNets) with dropout. Why were the Wide ResNets not trained with dropout? Is it because dropout can affect the Batch Normalization variance statistics? (During training, activations have more variance because dropout is on. In testing, they have less variance since dropout is off.)

---

> > > ### Comment · Area_Chair1 · 2018-11-25
> > > **Tuning on OOD samples is an issue**
> > >
> > > I think the whole evaluation should be done assuming that we don't know how it looks the OOD set. I mean, if you know how it looks, then it's just a typical toy problem of two class classification. I understand this is the way previous work address this problem, but it's time that this is changed.
> > >
> > > Liang et al. 2018 (ICLR 2018) made the assumption of validation OOD sets, but it is only for finding two hyper-parameters, T and epsilon. So, it is more or less OK as one can expect that the validation is not too much overfitted. Furthermore, there are a few papers on the problem at that time. Lee et al. 2018 (NIPS 2018) also used the validation OOD set, but it is for fining which layers are most effective on designing their feature-based score. So, it is more or less OK as one can expect that the validation is overfitted to the network architecture, not to the OOD dataset. Lee et al. 2018 (NIPS 2018) also show experimental results without assuming any external validation OOD sets at all. Compared to these prior works, the authors used validation OOD sets up-to-next level due to the nature of their approach (although they use the same number of validation samples as in the prior works). Namely. the authors run an optimizer searching some parameters over an infinite domain to fit some validation data, while prior works searches over some given small finite set. Hence, it is natural to expect that the testing performance of the authors' method would be highly dependent on the choice of validation.
> > >
> > > Table 6 is not convincing enough for the issue as the characteristics of TinyImageNet and LSUN can be similar as OOD. The performance drops under synthetic OOD validation indeed implies that the proposed method is highly dependent on the choice of validation. Lee et al. (ICLR 2018) indeed addressed the same issue and design a GAN to overcome the lack of knowledge of OOD (they still used OOD validation to tune a few hyper-parameters in their experiments though).
> > >
> > > For example, I think the setting that you split the same dataset (CIFAR or ImageNet) for 50% classes for In and 50% classes for Out is meaningful to see whether your method is indeed not overfitted to validation.
> > >
> > > If I miss something, please let me know.

---

> > > > ### Author Response · Authors · 2018-11-26
> > > > **Authors' response**
> > > >
> > > > Before going on with our answer, we would like to remark that Lee et al. (to appear in NIPS 2018) is concurrent work and was only a preprint at the time of ICLR submission (Sep. 27). Nevertheless, we did the effort to include their work in our comparison, and showed our method performs similar or better and has practical advantages.
> > > >
> > > > Secondly,  we would like to stress that the main point of our work is not to criticize or to do a survey against previous work, but to show that the BN running statistics contain useful information on the training dataset, that they are already pre-computed, and that it is a surprisingly robust measure for OOD detection. Hence, we feel the proposed method is a step in popularizing OOD detection, allowing for a plug and play metric that can work on already-trained models.
> > > >
> > > >
> > > > We completely agree with your observation that the true challenge lies in evaluating on unseen OOD datasets.
> > > > This is why (as opposed  to what is done in work previous to Lee et al. and ours) we also show results training only on a given  OOD dataset, and testing on all the others. Moreover, in the revised version, we have added results using fully unsupervised training, i.e., only using ID samples, with no knowledge of OOD samples at all nor need to generate adversarial samples.
> > > >
> > > > We do not agree, however, that our method suffers more from overfitting to the OOD validation set than previous and concurrent methods. Namely:
> > > > - Even though ODIN (Liang et al. 2018) uses only two parameters, the method's performance significantly drops when testing on unseen  OOD sample. For example, we show that even with only two parameters ODIN does not generalize well to synthetic datasets or to iSUN (see Table 7 of our manuscript, especially the AUROC metric, that does not depend on the chosen threshold);
> > > > - The work of Lee at al. (to appear in NIPS 2018) uses the validation OOD set to train the weights of the final OOD detector. This is clearly stated in the last sentence of page 4 in Lee et al. 2018:  "In our experiments, following similar strategies in [22], we choose the weight of each layer α_l by training a logistic regression detector using validation samples." Besides this, they also need to set the hyperparameters of the pre-processing step, so they do need to tune multiple parameters to the OOD.
> > > >
> > > > We also perform all evaluations on four backbones models, showing that the results obtained do not vary significantly for different backbones (differently than what we found for ODIN, for example). Moreover, in Figure 7 in the Appendix, we demonstrate that even when reducing the number of layers from which we extract the features and, consequently, the number of parameters, our method continues to show reasonable performance generalizing to unseen OOD datasets.
> > > >
> > > > We thank the AC for engaging in the review and look forward to more discussion,
> > > > Authors.

---

> > > ### Author Response · Authors · 2018-11-26
> > > **Authors' response**
> > >
> > > Just to clarify, as we discuss in the previous work section, the baseline work by Hendrycks et al. 2017 is only a threshold-based method; there is no (hyper)parameter to be tuned on (given the TNR @ 95% TPR metric). Notwithstanding, all state-of-the-art-methods do use OOD samples to tune the performance. This is what we do, but we also show how the method generalizes to unseen OOD datasets.
> > >
> > > We trained the WideResNet without dropout since we use exactly the same setup as Liang et al. 2018. They did not use dropout, as it is stated in their paper in page 3: "In addition, we evaluate the method on a Wide ResNet, with depth 28, width 10 (WRN-28-10) and dropout rate 0".
> > >
> > > Note that It was not our intention to disable the dropout because it might affect the BN statistics. As part of a more general analysis (that is beyond this paper), it might be interesting to see how dropout or other regularization schemes can affect the BN statistics.
> > >
> > > Thank you for pointing it out,
> > > Authors.

---

### Public Comment · (anonymous) · 2018-10-23
**Efficient approach, but still have some questions**

This paper presents a method to detect out-of-distribution (OOD) images. Experiments are done on cifar-10 and cifar-100 datasets, and the evaluation is done on standard OOD image datasets, according to ODIN.

The approach is promising because it can achieve high performance by using only single network, and it also needs only one forward pass to come up with OOD detection. Information that is taken into account is just existing information from batch normalization. This is very efficient.

Anyway, I have implemented this approach, but I observed that it is quite specific to the dataset. For example, I pick 1,000 images from cifar-10 test set as In-distribution, and a combination of 1,000 images from TinyImageNet (cropped) plus 1,000 images from gaussian noise images as OOD, to train logistic regression (or whatever the model is). I definitely come up with very high performance. But after that I let the model classify the training image, all of them are detected as Out-of-distribution. This is a problem.

We can overcome this problem very easy by adding 1,000 of training images as In-distribution to logistic regression training. But I come up with the question, is the model trained in the overfitting way? Does it hurt the generalization of model? For instance, TinyImageNet dataset contains dog images, which model have learnt from cifar-10 dataset, but this approach considers all of them as OOD.

Another question is that, can we train the model in this approach (OOD detection part) using fully unsupervised manner? Because, in my opinion, we cannot take into account all kind of OOD images that might be existing in the real world.

By the way, using the information from batch normalization in OOD detection really surprise me. This is simple but interesting. I hope this field of research will be helpful for an improvement in AI safety in near future.

---

> ### Author Response · Authors · 2018-10-24
> **Authors' response**
>
> Dear Engkarat Techapanurak,
>
> Thank you for your comment.
>
> Regarding your question on ID/OOD generalization ability of the logistic regression, we have done several experiments throughout the paper fitting on a few ID/OOD datasets and testing on unseen OOD datasets, that worked remarkably well, showing that this method is able to generalize to unseen OOD datasets. For the specific scenario you mentioned, when applying the OOD detector to the dataset used to train the model, we did this experiment and were able to detect  96% of the samples as ID without retraining the logistic regression or changing the threshold learned using the validation set. Did you use the code linked in the paper to obtain your results? If not, if you could share more information about your implementation, we could try to understand where the two implementations diverge. Again, we have not observed the effect you mentioned using our code.
>
> About the fact that TinyImageNet dataset contains dog images, we follow the benchmark setup as in ODIN paper, and there they also considered all images from this dataset as OOD. We agree that the next natural step for OOD detection should start considering semantic differences and overlaps between the classes of the ID/OOD datasets.
>
> Lastly, we are currently exploring unsupervised training and think this is a natural continuation to the method we have proposed.
>
> Bests,
> Authors.

---

> > ### Public Comment · (anonymous) · 2018-10-31
> > **Update about experiment on WRN-28-10**
> >
> > Dear Authors,
> >
> > Thank you for your reply.  And I am sorry for lack of information message.
> >
> > About the issue above, I did the experiment in different architecture of network containing simply 7 layers of CNN and ended with global average pooling. This network produces 85% accuracy on Cifar-10 test dataset, where wrn-28-10 got 96% accuracy. I observed that in some small architecture, low-order statistic between training and test dataset (seen/unseen by model) are quite different. I am not sure if it matters, and other kind of network is also in your consideration.
> >
> > But now I have tried on the same architecture with author which is WRN-28-10. This problem does not exist. It truly works remarkably well as authors mentioned.

---

> > > ### Author Response · Authors · 2018-11-05
> > > **Authors' response**
> > >
> > > We also considered in the paper the DenseNet and WRN40-4  (in the Appendix) without the architecture choice having a large effect on the performance of the proposed method. Regarding different architectures, we have been able to replicate the work (and get good results) in different architectures that were not included in the submitted paper (BN-NiN, for example), but we still consider that more investigation needs to be done in order to understand in which architectures/tasks the proposed scheme works well, as well as if there are other factors that might affect performance (such as the model performance for the task).
> > >
> > > Bests,
> > > Authors.

---

### Public Comment · ~ioui_wc1 · 2018-11-02
**prefermance when In-distribution are partially overlapping with the Out-distributions?**

Hi, very insteresting paper!

I am wondering that if the in-distribution dataset and out-distribution datasets share some classes, what will happen?
Will those shared classes be kept as in-distribution? Because in real industrial environment, it is rarely that the testing dataset distribution follows the train dataset distribution very well.

---

> ### Author Response · Authors · 2018-11-05
> **Authors' response**
>
> Hi,
>
> Thank you for your feedback!
>
> As was done in previous approaches, we did not account for overlaps between the classes of the ID/OOD datasets. We do believe that such discussion merits a more thorough investigation, especially looking into what constitutes an OOD-sample.  For instance, if it is a sample belonging to a training class but in a different domain (e.g. indoor vs outdoor or day vs night), or if it is a sample from the same domain but of a different class (e.g. CIFAR80 vs CIFAR20). Also, bear in mind that instead of a binary classification problem, one could use the score of the logistic regressor as an OOD measure (e.g., as a probability of being out-of-distribution), thus allowing for different tradeoffs when detecting the OOD samples depending on the application.
>
> Thanks,
>
> Authors.

---

### Author Response · Authors · 2018-11-19
**Reviewed version**

We thank all the reviewers, which have raised valid questions and concerns about our paper. We reviewed it fully and amended it wherever possible. In particular, we added/changed the following contents:
1. Main paper:
  a. Added some short discussion about the proposed method weakness in section 3.2;
  b. Compacted both section 2 and 3 (addressing Reviewer 3 concerns);
  c. Changed some captions to make explicit what is being calculated and which OOD datasets have been fitted;
  d. Divided the experiment section into two: pairwise fitting  vs. non-pairwise fitting;
  e. Replaced Table 6, showing how the choice of OOD dataset used for training the logistic regressor impacts the
      performance of the method. More importantly, we’ve also demonstrated that even when training the OOD classifier in
      an unsupervised manner, the proposed method still exhibits competitive performance.
2. Appendix:
  a. B.2: added missing Table 9 with the error rate of WRN-40-4;
  b. B.4: added Table 10, showing extra results using SVHN as ID;
  c. B.7: added Table 12, showing extra results using different TPR levels (97% and 99%).

Due to our aforementioned modifications, we had to do minor modifications throughout the paper to fit it in 10 pages.

Bests,
Authors.

---

### Meta-Review · Area_Chair1 · 2018-12-12
**Interesting simple idea, but limited novelty and unfair experimental setups**

**Confidence:** 5
**Recommendation:** Reject

**Metareview:**

The paper proposes a simple method for detecting out-of-distribution samples. The authors' major finding is that mean and standard deviation within feature maps can be used as an input for classifying out-of-distribution (OOD) samples. The proposed method is simple and practical.

The reviewers and AC note the following potential weaknesses: (1) limited novelty and somewhat ad-hoc approach, i.e., it is not too surprising to expect that such statistics can be useful for the purpose. Some theoretical justification might help. (2) arguable experimental settings, i.e., the performance highly varies depending on validation (even in the revised draft), and sometimes irrationally good. It also depends on the choice of classifier.

For (2), I think the whole evaluation should be done assuming that we don't know how it looks the OOD set. Under the setting, the authors should compare the proposed method and existing ones for fair comparisons. AC understands the authors follows the same experimental settings of some previous work addressing this problem, but it's time that this is changed. Indeed, a recent paper by Lee at al. 2018 considers such a setting for detecting more general types of abnormal samples including OOD.

In overall, the proposed idea is simple and easy to use. However, AC decided that the authors need more significant works to publish the work.

---

> ### Public Comment · (anonymous) · 2019-01-21
> **How is Lee at al. (NIPS 2018) settings for experiments the right one?**
>
> As already pointed out in one of the author's responses (https://openreview.net/forum?id=rkgpCoRctm&noteId=ryx10nht07), Lee at al. (NIPS 2018) uses OOD dataset to train the logistic regressor and also to finetune other hyperparameters in preprocessing. In that case, how is their setting for experiments is any different from the current paper? Or am I totally missing something?